# The Moderating Role of Self-Control and Financial Strain in the Relation between Exposure to the Food Environment and Obesity: The GLOBE Study

**DOI:** 10.3390/ijerph16040674

**Published:** 2019-02-25

**Authors:** Joreintje D. Mackenbach, Marielle A. Beenackers, J. Mark Noordzij, Joost Oude Groeniger, Jeroen Lakerveld, Frank J. van Lenthe

**Affiliations:** 1Department of Epidemiology and Biostatistics, VU University Medical Center, Amsterdam Public Health Research Institute, 1007 MB Amsterdam, The Netherlands; je.lakerveld@vumc.nl; 2Department of Public Health, Erasmus University Medical Center, 3000 CA Rotterdam, The Netherlands; m.beenackers@erasmusmc.nl (M.A.B.); j.m.noordzij@erasmusmc.nl (J.M.N.); j.oudegroeniger@erasmusmc.nl (J.O.G.); f.vanlenthe@erasmusmc.nl (F.J.v.L.); 3Department of Public Administration and Sociology, Erasmus University, 3000 DR Rotterdam, The Netherlands; 4Julius Center for Health Sciences and Primary Care, University Medical Center Utrecht, 3584 CX Utrecht, The Netherlands; 5Faculty of Geosciences, Department of Human Geography and Spatial Planning, Utrecht University, 3508 TC Utrecht, The Netherlands

**Keywords:** body weight, cognitive bandwidth, food environment, GIS, interaction

## Abstract

Low self-control and financial strain may limit individuals’ capacity to resist temptations in the local food environment. We investigated the moderating role of self-control and financial strain in the relation between the food environment and higher body weight. We used data from 2812 Dutch adults who participated in the population-based GLOBE study in 2014. Participants’ home addresses and the location of food retailers in 2013 were mapped using GIS. The density of fast food retailers and the totality of food retailers in Euclidean buffers of 250, 400 and 800 m around the home were linked to body mass index and overweight status. A higher density of fast food outlets (B (95% confidence interval (CI)) = −0.04 (−0.07; −0.01)) and the totality of food outlets (B (95% CI) = −0.01 (−0.01; −0.00)) were associated with a lower body mass index. Stratification showed that associations were strongest for those experiencing low self-control or great financial strain. For example, every additional fast food outlet was associated with a 0.17 point lower BMI in those with great financial strain, while not significantly associated with BMI in those with no financial strain. In conclusion, we did find support for a moderating role of self-control and financial strain, but associations between the food environment and weight status were not in the expected direction.

## 1. Introduction

Local food environments are part of the complex web of factors that influence food choices and obesity [1,2,3,4,5,6]. Ubiquitous access to convenient and inexpensive food has changed normative eating behaviour, with more people snacking and eating out-of-home [7]. Living in an ‘obesogenic environment’ with high accessibility to energy-dense and ultra-processed foods and repeated exposure to food cues such as advertisements, smells and promotions increases the likelihood that individuals indulge themselves [8]. These temptations—i.e., the omnipresence of tempting food cues —distract from long-term health objectives including healthy diet and body weight. 

Tempting cues are considered especially influential in people with a burdened mental capacity, whose ability to regulate the impact of temptations on behaviour is limited [9,10]. Self-control refers to the capacity to override and replace a dominant response (e.g., give in to a tempting food cue) with a response more in line with longer term goals (e.g., sticking to a dietary plan) [11,12,13]. Self-control is facilitated by self-regulation. Although a topic of debate [14], self-regulation is hypothesized to be functionally confined [15]; that is, individuals that used self-control on one task will function more poorly on subsequent tasks that also demand self-control [15]. This temporary deficit in the skills needed to effectively manage thoughts, feelings and actions is known as a state of ego depletion [15]. Procedures or tasks that occupy attention are presumed to create cognitive load [16] (or reduce ‘cognitive bandwidth’ [17]). With reduced cognitive bandwidth, fewer processing resources are available for other information. Research suggests that both ego depletion and increased cognitive load undermine performance on tasks that require conscious, controlled and complex cognitive processes [18,19,20], such as resisting temptations in the food environment.

Arguably amongst the most important sources of cognitive load is financial strain: a constant stressor that forces daily difficult financial decision making on basic matters such as food and clothing. Previous studies have shown that relentless stress and a feeling of lack of control negatively impact health [21,22]. These results are in line with the ‘scarcity theory’ [17,23], which suggests that dealing with scarcity (such as scarcity of money) takes up cognitive bandwidth that could otherwise be used to resist temptations [17]. 

In contemporary food environments, food is available anytime and anywhere, and the consumption of high-energy and ultra-processed foods is strongly promoted [24]. In such ‘obesogenic’ environments, food choices that are made while experiencing great financial strain or low perceived self-control are expected to be less healthy since the easy choice is often an unhealthy choice. However, research testing this hypothesis is still in its infancy. One study conducted among Canadian adults demonstrated that a relationship between a high level of mastery (a construct closely related to self-control) and lower metabolic risk (an outcome closely related to dietary intake) was most apparent in environments with higher fast-food exposure [25]. A cross-European study among adolescents showed that, although easy access to unhealthy food products was associated with higher consumption, the effect was attenuated when self-regulation strategies were used [26]. 

Building on the evidence that both low self-control and high financial strain are linked to unhealthier lifestyle behaviours and increased body weight [27,28,29,30,31,32], we aimed to examine whether associations between the local food environment and weight status differed across levels of self-control and financial strain. We hypothesized that unhealthier food environments are associated with higher body weight in those experiencing low self-control or great financial strain, but not or only weakly in individuals experiencing high self-control or no financial strain.

## 2. Materials and Methods 

### 2.1. Data

We used cross-sectional data from the longitudinal Dutch population-based GLOBE study. Participants were adults (25–75 years) living in Eindhoven and surrounding cities. The *N* = 2812 participants (45.5% response rate) were representative of the source population (residents of Eindhoven and surroundings aged 25–75 years) in terms of health-related behaviours [33]. We used data from 2014, when participants were asked to complete a questionnaire. More detailed information on the objectives, study design and data collection of the Dutch GLOBE study can be found elsewhere [33]. The use of personal data in the GLOBE study follows the Dutch Personal Data Protection Act and the Municipal Database Act and has been registered with the Dutch Data Protection Authority (number 1248943).

### 2.2. Measures

#### 2.2.1. Outcome Variables

BMI was calculated using self-reported height and weight. Weight status was classified into normal weight (18 kg/m^2^ > BMI < 25 kg/m^2^), overweight (25 kg/m^2^ ≥ BMI < 30 kg/m^2^) and obesity (BMI ≥ 30 kg/m^2^). 

#### 2.2.2. Exposure Variables

Unhealthier food environments were defined as local neighbourhoods with (1) more opportunities to buy fast foods or takeaway meals, and (2) a higher totality of food retailers. Participants’ home addresses were mapped using a geographic information system (GIS-ArcGIS 10, ESRI, Redlands, CA, USA). The location of food retailers dating from 2013 was also mapped in this GIS. Food retailer data was obtained from Locatus [34], a commercial company that performs yearly audits on all retailers in the Netherlands. We considered all fast food and take away outlets available in the Locatus database as fast food restaurants. In accordance with the classification of Lake et al. [35], fast food and take away outlets were defined as outlets serving hot food ordered and paid for at the till, foods cooked in bulk in advance and providing no or minimal table service (chain and non-chain fast food restaurants, take away and delivery outlets, grillrooms and kebab shops). The totality of food outlets consisted of the sum of all types of food outlets (e.g., supermarkets, greengrocers, convenience stores, bars, restaurants, etc.). 

We calculated the density (number per km^2^) of fast food outlets and the totality of food outlets in Euclidean buffers of 400 m (representing a walk of approximately 5 minutes) and conducted sensitivity analyses with 250 m and 800 m buffer sizes (also walkable and cyclable distances). 

Our exposure variables were non-normally distributed, but transformations did not improve the distribution. We therefore treated the untransformed densities as continuous variables to facilitate the interpretation of the main effects and effect modification (i.e., a 1-unit increase in the densities represents an increase of 1 food outlet per km^2^ in the 400 m buffer), and additionally conducted sensitivity analyses with density categorized into quartiles. 

#### 2.2.3. Effect Modifiers

Financial strain was assessed by two questions addressing (1) whether participants could make ends meet considering their monthly household income and (2) whether they had experienced any financial difficulties in paying bills for food, rent, electricity and so forth during the preceding year. Those two items were combined into having no, some or great financial strain: ‘no financial strain’ if they could make ends meet fairly easy or easy, or if they experienced no financial difficulties in the preceding year; ‘some financial strain’ if they could make ends meet with some difficulty or if they experienced some financial difficulties in the preceding year; and ‘great financial strain’ if they had great difficulty making ends meet or if they experienced great financial difficulties in the preceding year. 

Self-control was measured using the Brief Self-Control Scale [36], a widely used instrument for measuring general trait self-control. Participants rated each of the 13 items (e.g., “I am good at resisting temptation”) on a 5-point scale ranging from 1 (not at all like me) to 5 (very much like me). This scale was internally consistent (Cronbach’s alpha = 0.78) and could range from 13 to 65. Self-control was divided into tertiles for stratified analyses. 

#### 2.2.4. Confounding Variables

Age (continuous in years), sex (male, female), country of birth (Netherlands, other), employment status (full-time employed, part-time employed, unemployed, retired, homemaker, other), length of residency (continuous in years), household equivalent income and educational level were selected a priori as potential confounders. Household equivalent income was measured as the level of household income per month divided by the square root of the number of people living from this income and divided into 5 categories: 1—< €1000/month, 2—€1000–€1500/month, 3—€1500–€2 000/month, 4—€2000–2500/month and 5—> €2500/month. Highest attained educational level was classified according to the International Standard Classification of Education (ISCED): 1—primary education (ISCED 0–1), 2—lower secondary education (ISCED 2), 3—upper secondary education (ISCED 3–4), 4—tertiary education (ISCED 5–7).

### 2.3. Analyses

Assuming that missing data were missing at random, they were handled using multiple imputation for inferential statistics. The pattern of missingness showed that 40 (<1%) participants had missing data on BMI and 414 (15%) participants had missing data on any covariate or outcome measure. The chained equations function (with predictive mean matching for continuous variables, logistic regression for dichotomous variables, multinomial logistic regression for categorical variables and ordered logistic regression for ordered variables) was used to perform multiple imputation analyses. Based on the overall percentage missing values (13%) we generated 13 multiple imputed datasets, as recommended by Rubin [37] and Bodner [38]. All variables described above were used in the imputation model and smoking, alcohol consumption, general health and wellbeing were treated as auxiliary variables. 

Pooled results from the 13 imputed datasets were used in the regression models and non-imputed data were used to perform the descriptive analysis on the characteristics of the study population and their food environment.

Linear and multinomial logistic regression analyses were performed with body mass index and weight status as outcome variables, adjusted for confounders. Unadjusted and partially adjusted models were also explored, but given the similar results, only fully adjusted models are presented. Individuals with underweight (BMI < 18 kg/m^2^; *N* = 30) were excluded from the analyses with weight status as outcome variable. Statistical significance was interpreted using 95% confidence intervals. Effect modification was assessed by stratifying the analyses by level of financial strain and self-control. Sensitivity analyses were conducted with density of fast food and the totality of food retailers in 250 m and 800 m buffers, and with densities divided in quartiles. All analyses were weighted by respondent-level sample weights to account for the sampling strategy used within the GLOBE study. All analyses were carried out in STATA, version 14.1 (StataCorp, College Station, TX, USA). 

## 3. Results

Descriptive statistics are presented in Table 1. The mean age of the sample was 49 years, 45% were male and the average length of residency in the neighbourhood was 16 years. The average density of the totality of food outlets in a 400 m buffer was 17 outlets per km^2^, and the density of fast food outlets was 4.1 outlets per km^2^. 

Table 2 shows that one extra fast food outlet per km^2^ in a 400 m buffer was associated with a 0.04 (95% CI = −0.07; −0.01) lower body mass index. Similarly, one extra food outlet per km^2^ was associated with a 0.01 (95% CI = −0.01; 0.00) lower body mass index. The density of fast food outlets or the totality of food outlets were not significantly associated with weight status, although the direction of the coefficients suggests a negative association as well. Sensitivity analyses using 250 m and 800 m buffers instead of 400 m buffers resulted in comparable effect sizes in the same direction (Appendix A). Sensitivity analyses using quartiles of densities showed similar negative associations with body weight, with the strongest effect sizes in quartile 4 (highest density; Appendix A). 

Table 3 displays the associations between measures of the food environment and body mass index stratified by levels of financial strain and self-control. A higher density of fast food outlets or totality of food outlets was mainly associated with a lower body mass index in individuals experiencing great financial strain (B = −0.17, 95% CI = −0.33; −0.02) and low self-control (B = −0.09, 95% CI = −0.15; −0.02). 

Similar patterns were found with weight status as the outcome (Table 4), although only the associations between food environment measures and overweight status in those experiencing great financial strain reached statistical significance. For example, one extra fast food outlet per km^2^ in a 400 m buffer was associated with a 0.90-times lower relative risk of overweight and obesity. 

## 4. Discussion

We hypothesized that individuals with reduced cognitive bandwidth, operationalized as low perceived self-control and greater financial stress, are more vulnerable to unhealthy temptations in the food environment. We observed that associations between the food environment and body weight were stronger in those experiencing low self-control or great financial strain, but the direction of these associations was unexpected; that is, the density of fast food outlets and the totality of food outlets in the local neighbourhood were negatively related to body mass index and odds of being overweight and obese, and particularly among those with low levels of self-control and greater financial strain.

In the main analysis, every additional fast food outlet per squared kilometre was associated with a 0.04-point lower BMI. For a person of 1.80 m and 80 kg, this translates to 0.13 kg less body weight. Associations with overweight and obesity were in the same direction and with similar small effect sizes but did not reach statistical significance. It is puzzling that, regardless the size of the local area considered, a higher availability of fast food outlets was associated with lower body weight. Similar but smaller associations were observed when analysing the totality of food outlets. This may suggest that the totality of food outlets reflects opportunities for buying healthier and unhealthier foods, more so than representing an overabundance of food outlets. 

Our study is not the first to find unexpected results between the fast food environment and indicators of body weight: systematic reviews describe conflicting results in the literature [39,40,41,42]. The expected mechanism of this association is through dietary behaviour. Even though we did not have such data available for the present study, unexpected results in previous literature and in the current study are not likely to be attributable to the lack of these data: fast food is typically nutrient-poor, calorie-dense and high in fat, sugar and salt, and consumption at fast food restaurants is associated with higher energy intake and greater obesity risk [43,44]. The inconsistencies could thus be attributable to misclassification in the exposure variables: the missing link between the availability of fast food outlets and use of these outlets. A recent European study demonstrated that while access to fast food outlets in the home neighbourhood was not directly linked to fast food consumption or obesity, access to fast food outlets was associated with perceived availability and use of fast food outlets, and this was in turn associated with greater reported fast food consumption and unhealthier weight status [45]. In addition, a recent study demonstrated that exposure to food outlets in the residential neighbourhood was not representative of the overall foodscape exposure [46], and another study showed that the work and commuting environment also contribute substantially to the exposure to food outlets [47]. As such, the density of food outlets in the residential neighbourhood may not reflect the total exposure to food outlets individuals encounter in their daily lives. Studies using global positioning systems (GPS) may provide insight into what fast food outlets individuals are exposed to, and which of these outlets they visit [48]. In addition, it could be speculated that a higher density of food outlets in the residential neighbourhood is just a reflection of greater ‘land use mix’ (i.e., an area with multiple types of destinations, including food retailers), which previously has been linked to walking and lower rates of obesity [49,50]. We hypothesized that the food environment would be associated with higher body weight via dietary behaviour (energy intake) but did not investigate associations with energy expenditure: it may be that having more food outlets within a walkable and cyclable distance is in fact associated with more physical activity. 

A previous study in the same cohort showed that low self-control and high financial strain are linked to unhealthier lifestyle behaviours and increased body weight [32], and we found evidence for an interaction between these social cognitive factors and the food environment. It is uncertain whether the influence of the food environment on obesity is always stronger in those with low self-control and high financial strain (i.e., would similar moderating effects be present had we observed a positive food environment–body weight association?), or whether this moderation is limited to the negative food environment–body weight association as observed in the present study. Future studies, preferably with more precise data on what temptations in the food environment individuals are actually exposed to, should investigate whether experiencing low self-control and great financial strain take up cognitive bandwidth that could otherwise be used to resist unhealthy temptations in the food environment, as proposed by the scarcity theory [17,23].

### Strengths and Limitations

This is the first study to investigate the potential moderating role of reduced cognitive bandwidth on the association between the local food environment and body weight. Although the study design (using questionnaires) will have excluded participants with low literacy and low mastery of the Dutch language, our sample was representative of the source population (residents of Eindhoven and surroundings aged 25–75 years) in terms of health-related behaviours [33]. Another strength is the linkage of two objective measures of exposure to the food environment with weight status. However, the results of this study should be interpreted in the light of its limitations. First, we used self-reported height and weight, and reporting bias by those with short height or high weight may have resulted in an underestimation of BMI and a potential misclassification of weight status [51]. This may have attenuated the associations under study. Second, exposure misclassification may have arisen because our measures of the food environment may not accurately reflect where participants bought food and where they experienced temptation to buy unhealthy food. We accounted for exposure misclassification due to area definitions by using different ego-centred buffers, but the density of fast food outlets and the totality of food outlets in the residential neighbourhood may not have captured the obesogenic aspects of the food environment well enough. Third, residual confounding in food environment research has been shown to result in null or weak findings [52], and this may be applicable in this study as well. Fourth, our study was unable to capture the complex system of interacting factors of influence on obesity [2], despite our investigation of the interaction between two environmental-level and two individual-level factors. This limitation includes the cross-sectional design, which restricted our opportunities of disentangling selection, causation and time-lag effects of food environments on body weight, and the lack of data on the hypothesized mediating variable dietary intake.

## 5. Conclusions

We did find support for a moderating role of self-control and financial strain, but associations between the food environment and weight status were not in the expected direction. Temptations in the food environment may be better captured by other measures than the density of (fast) food outlets, but interactions between unhealthy food environments and cognitive resources such as self-control should be further explored. 

## Figures and Tables

**Table 1 ijerph-16-00674-t001:** Sample characteristics.

Measures	Total Sample (*N* = 2812)
Age (mean (sd) years)	48.8 (14.9)
Gender (*N* (%) male)	1332 (44.8%)
Household equivalent income	
≤1000€ per month	351 (14.3%)
1000–1500€ per month	508 (20.7 %)
1500–2000€ per month	608 (24.8%)
2000–2500€ per month	690 (28.1%)
>2500€ per month	298 (12.1%)
Educational level	
Low (ISCED 0–2; *N* (%))	761 (25.8%)
Medium (ISCED 3–4; *N* (%))	740 (25.1%)
High (ISCED 5–8; *N* (%))	1448 (49.1%)
Country of birth (*N* (% Netherlands))	2480 (88.5%)
Employment status	
Employed (*N* (%))	1860 (63.6%)
Unemployed (*N* (%))	234 (20.4%)
Retired (*N* (%)	598 (8.0%)
Non-employed (*N* (%))	232 (7.9%)
Children living in the household	
No (*N* (%))	1714 (61.8%)
Yes (*N* (%))	1061 (38.2%
Length of residency in the current neighbourhood (mean (sd) years)	16.3 (13.4)
Body mass index (mean (sd) kg/m^2^)	25.6 (4.9)
Overweight (% ≥ 25 kg/m^2^)	34.4%
Obesity (% ≥ 30 kg/m^2^)	18.1%
Self-control (mean (sd) score)	44.1 (6.8)
Financial strain	
No financial strain (*N* (%))	1864 (68.1%)
Some financial strain (*N* (%))	689 (25.0%)
Great financial strain (*N* (%))	208 (6.9%)
Density of fast food outlets in a 400 m buffer (mean (sd) count per km^2^)	4.1 (4.8)
Density of all food outlets in a 400 m buffer (mean (sd) count per km^2^)	17.0 (27.1)

**Table 2 ijerph-16-00674-t002:** Associations between measures of the food environment and body weight.

Exposure Variables	Body Mass Index	Weight Status		
		Normal Weight	Overweight	Obesity
	B (95% CI)	RRR (95% CI)	RRR (95% CI)	RRR (95% CI)
Density of fast food outlets in a 400 m buffer	**−0.04** (−0.07; −0.01)	Ref.	0.99 (0.97; 1.01)	0.97 (0.94; 1.00)
Density of all food outlets in a 400 m buffer	**−0.01** (−0.01; −0.00)	Ref.	1.00 (0.99; 1.00)	1.00 (0.99; 1.00)

Note: B = regression coefficient. RRR = relative risk ratio. CI = confidence interval. Ref. = reference category. All analyses are adjusted for age, gender, education, children in the household, household equivalent income, employment status, country of birth and length of residency. ‘Normal weight’ was the reference category in the multinomial logistic regression analysis with weight status as dependent variable. Bold values represent statistically significant associations as defined by the 95% confidence interval.

**Table 3 ijerph-16-00674-t003:** Associations between measures of the food environment in a 400m buffer and body mass index stratified by financial strain and self-control.

**Exposure Variables**	**Body Mass Index**		
	**High Self-control (*N* = 725)**	**Medium Self-control (*N* = 1059)**	**Low Self-control (*N* = 1012)**
	**B (95% CI)**	**B (95% CI)**	**B (95% CI)**
Density of fast food outlets	−0.01 (−0.07; 0.05)	−0.03 (−0.08; 0.03)	**−0.09** (−0.15; −0.02)
Density of all food outlets	−0.00 (−0.01; 0.01)	−0.01 (−0.01; 0.00)	**−0.01** (−0.02; −0.00)
	**No Financial Strain (*N* = 1864)**	**Some Financial Strain (*N* = 689)**	**Great Financial Strain (*N* = 208)**
	**B (95% CI)**	**B (95% CI)**	**B (95% CI)**
Density of fast food outlets	−0.04 (−0.08; 0.00)	−0.02 (−0.10; 0.06)	**−0.17** (−0.33; −0.02)
Density of all food outlets	**−0.01** (−0.01; −0.00)	−0.00 (−0.01; 0.01)	**−0.03** (−0.05; −0.00)

Note: B = Regression coefficient. CI = Confidence Interval. All analyses are adjusted for age, gender, education, children in the household, household equivalent income, employment status, country of birth and length of residency. Bold values represent statistically significant associations as defined by the 95% confidence interval.

**Table 4 ijerph-16-00674-t004:** Associations between measures of the food environment in a 400 m buffer and weight status stratified by financial strain and self-control.

**Exposure Variables**	**Weight Status**						
	**Normal Weight**	**Overweight**			**Obesity**		
		**No Financial Strain**	**Some Financial Strain**	**Great Financial Strain**	**No Financial Strain**	**Some Financial Strain**	**Great Financial Strain**
		RRR (95% CI)	RRR (95% CI)	RRR (95% CI)	RRR (95% CI)	RRR (95% CI)	RRR (95% CI)
Density of fast food outlets	Ref.	1.00 (0.97; 1.02)	1.00 (0.96; 1.04)	**0.90** (0.83; 0.96)	0.98 (0.94; 1.02)	0.96 (0.90; 1.02)	**0.90** (0.81; 1.00)
Density of all food outlets	Ref.	1.00 (0.99; 1.00)	1.00 (0.99; 1.00)	**0.98** (0.96; 0.99)	1.00 (0.99; 1.01)	0.99 (0.98; 1.00)	**0.98** (0.97; 1.00)
	**Normal Weight**	**Overweight**			**Obesity**		
		**High Self-Control**	**Medium Self-Control**	**Low Self-Control**	**High Self-Control**	**Medium Self-Control**	**Low Self-Control**
		RRR (95% CI)	RRR (95% CI)	RRR (95% CI)	RRR (95% CI)	RRR (95% CI)	RRR (95% CI)
Density of fast food outlets	Ref.	1.00 (0.96; 1.04)	0.99 (0.95; 1.02)	1.00 (0.97; 1.03)	1.01 (0.94; 1.08)	0.98 (0.92; 1.04)	**0.94** (0.90; 0.99)
Density of all food outlets	Ref.	1.00 (0.99; 1.00)	1.00 (0.99; 1.00)	1.00 (0.99; 1.00)	1.00 (0.99; 1.01)	1.00 (0.99; 1.01)	0.99 (0.98; 1.00)

Note: RRR = relative risk ratio. CI = confidence interval. Ref. = reference category. All analyses are adjusted for age, gender, education, children in the household, household equivalent income, employment status, country of birth and length of residency. ‘Normal weight’ was the reference category in the multinomial logistic regression analysis with weight status as dependent variable. Bold values represent statistically significant associations as defined by the 95% confidence interval.

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
