# Peer review of "The Moderating Role of Self-Control and Financial Strain in the Relation between Exposure to the Food Environment and Obesity: The GLOBE Study"

_ijerph, 2019, doi:10.3390/ijerph16040674_

Round 1

Reviewer 1 Report

Abstract: more detailed results should be presented in the abstract. Methods: a simple introduction on the study design and data collection shoud be added.food deliver via on-line or mobile order shoud be taken into consideration in the food environment variables.Culture, and religous should be taken into consideration into the confounding variables.the possible influence of low response rate on the results and its limitation  should be discussion.

Author Response

Reviewer 1

Comments and Suggestions for Authors

Abstract: more detailed results should be presented in the abstract.

>> Taking into account the word count limit (max. 200 words) of the journal, we have now exchanged details about the methodology for details about the results. The abstract now reads as follows:

“Low self-control and financial strain may limit individuals’ capacity to resist temptations in the local food environment. We investigated the moderating role of self-control and financial strain in the relation between the food environment and higher body weight. We used data from 2812 Dutch adults, who participated in the population-based GLOBE study in 2014. Participants’ home addresses and the location of food retailers in 2013 were mapped using GIS. The density of fast food retailers and the totality of food retailers in Euclidean buffers of 250, 400 and 800m around the home were linked to body mass index and overweight status. Higher density of fast food outlets (B(95%CI)=-0.04(-0.07;-0.01)) and the totality of food outlets (B(95%CI)=-0.01(-0.01;-0.00)) was associated with a lower body mass index. Stratification showed that associations were strongest for those experiencing low self-control or great financial strain. For example, every additional fast food outlet was associated with a 0.17 point lower BMI in those with great financial strain, while not significantly associated with BMI in those with no financial strain. In conclusion, we did find support for a moderating role of self-control and financial strain, but associations between the food environment and weight status were not in the expected direction.”

Methods: a simple introduction on the study design and data collection shoud be added.

>> We have simplified the description of the study design and data collection and provide additional details. Section 2.1 Data now reads as follows:

“We used cross-sectional data from the longitudinal Dutch population-based GLOBE study. Participants were adults (25-75y) living in Eindhoven and surrounding cities. The N=2 812 participants (45.5% response rate) were representative of the source population (residents of Eindhoven and surroundings aged 25-75 years) in terms of health-related behaviours[33]. We used data from 2014, when participants were asked to complete a questionnaire. More detailed information on the objectives, study design and data collection of the Dutch GLOBE study can be found elsewhere[33]. The use of personal data in the GLOBE study follows the Dutch Personal Data Protection Act and the Municipal Database Act and has been registered with the Dutch Data Protection Authority (number 1248943).”‍

Food deliver via on-line or mobile order shoud be taken into consideration in the food environment variables.

>> We agree with the reviewer that food delivery services are an important aspect of the food environment, which is why we included take away/delivery outlets in our food environment variables. We did not have information on the delivery radius of the food delivery services, which is why we did not include food delivery outlets outside the pre-defined buffers of 250, 400 and 800 meters.

Culture, and religous should be taken into consideration into the confounding variables.

>>We have carefully considered the reviewer’s suggestion to include culture and religion as confounders in our models. A confounder is a variable that influences both the dependent variable and the independent variable. While culture is likely to be related to body weight, a relation between religion and body weight is not established, and culture and religion are unlikely to influence the number of fast food or other outlets. We are confident that we have captured the most important individual level confounders by including age, sex, country of birth (which may be seen as a cultural factor as well), employment status, length of residency, household equivalent income and educational level in the models.

The possible influence of low response rate on the results and its limitation  should be discussion.

>> This study had a response rate of 45.5%, which may be seen as a low or a reasonable response rate, depending on the field of study. What is more important is the representativeness of the study population. Although our Dutch questionnaire will have excluded those with low literacy and unable to read Dutch, our sample was representative of the source population in terms of health-related behaviours, as described in the Discussion section: “Although the study design (using questionnaires) will have excluded participants with low literacy and low mastery of the Dutch language, our sample was representative of the source population (residents of Eindhoven and surroundings aged 25-75 years) in terms of health-related behaviours[33].”

We have added this information to the Methods section so that this becomes clear to the readers earlier on: “The N=2 812 participants (45.5% response rate) were representative of the source population (residents of Eindhoven and surroundings aged 25-75 years) in terms of health-related behaviours[33].”

Reviewer 2 Report

Manuscript jerph-434833: Reviewer’s Comment:

This is quite an interesting paper. The introduction is quite good. The description of the methods and materials was well done. Authors did a nice job defining the variables and how each was measured. I think the data analysis presentation is good. The results presentation is straightforward and simple to follow. Discussion of the results is quite strong.

But the paper has a few typo/ grammatical errors that authors should address. 

1)      For example, the first sentence beginning on line 206 is not very clear.  “Authors We hypothesized that………….”   Please consider deleting “Authors”

2)      On lines 122-123, authors stated “…………. or if they experienced large financial difficulties in the preceding year. What about saying “or if they experienced enormous financial difficulties in the preceding year. Or please look for a better word to replace “large”

3)      One line 109, authors stated “Our independent variables were randomly distributed……” Is independent variable the same as exposure variables? If so, they should consider being consistent in the use of term.  If authors want to use the term “exposure variables” instead of independent variables, I think authors should maintain it all through the manuscript.

4)      The sentence beginning from line 39 to 41 sounds complete. “High accessibility of high-energy and ultra-processed foods……………………………………………………………., increase the likelihood that individuals indulge themselves [8].” This sentence is not very clear. Authors should consider revising the sentence.

Author Response

Comments and Suggestions for Authors

This is quite an interesting paper. The introduction is quite good. The description of the methods and materials was well done. Authors did a nice job defining the variables and how each was measured. I think the data analysis presentation is good. The results presentation is straightforward and simple to follow. Discussion of the results is quite strong.

>> We thank the reviewer for their positive feedback on our manuscript.

But the paper has a few typo/ grammatical errors that authors should address. 

1)      For example, the first sentence beginning on line 206 is not very clear.  “Authors We hypothesized that………….”   Please consider deleting “Authors”

>> We have deleted ‘Authors’ as suggested.

2)      On lines 122-123, authors stated “…………. or if they experienced large financial difficulties in the preceding year. What about saying “or if they experienced enormous financial difficulties in the preceding year. Or please look for a better word to replace “large”

>> We have replaced ‘large’ by ‘great’.

3)      One line 109, authors stated “Our independent variables were randomly distributed……” Is independent variable the same as exposure variables? If so, they should consider being consistent in the use of term.  If authors want to use the term “exposure variables” instead of independent variables, I think authors should maintain it all through the manuscript.

>> We now consistently refer to ‘exposure variables’ throughout the manuscript.

4)      The sentence beginning from line 39 to 41 sounds complete. “High accessibility of high-energy and ultra-processed foods……………………………………………………………., increase the likelihood that individuals indulge themselves [8].” This sentence is not very clear. Authors should consider revising the sentence.

>> We have revised this sentence as follows: “Living in an ‘obesogenic environment’, with high accessibility to energy-dense and ultra-processed foods, and repeated exposure to food cues such as advertisements, smells and promotions, increase the likelihood that individuals indulge themselves[8].”
